# Luminescent hyperbolic metasurfaces

J.S.T. Smalley[1], F. Vallini[1], S.A. Montoya[1], L. Ferrari[2], S. Shahin[1], C.T. Riley[3], B. Kanté[1], E.E. Fullerton[1,3], Z. Liu[1] & Y. Fainman[1]

When engineered on scales much smaller than the operating wavelength, metal-semiconductor nanostructures exhibit properties unobtainable in nature. Namely, a uniaxial optical metamaterial described by a hyperbolic dispersion relation can simultaneously behave as a reflective metal and an absorptive or emissive semiconductor for electromagnetic waves with orthogonal linear polarization states. Using an unconventional multilayer architecture, we demonstrate luminescent hyperbolic metasurfaces, wherein distributed semiconducting quantum wells display extreme absorption and emission polarization anisotropy. Through normally incident micro-photoluminescence measurements, we observe absorption anisotropies greater than a factor of 10 and degree-of-linear polarization of emission $>0.9$. We observe the modification of emission spectra and, by incorporating wavelength-scale gratings, show a controlled reduction of polarization anisotropy. We verify hyperbolic dispersion with numerical simulations that model the metasurface as a composite nanoscale structure and according to the effective medium approximation. Finally, we experimentally demonstrate $>350\%$ emission intensity enhancement relative to the bare semiconducting quantum wells.

[1] Department of Electrical and Computer Engineering, University of California, San Diego, 9500 Gilman Drive, Mail code 0407, La Jolla, California 92093, USA. [2] Material Science and Engineering Program, University of California, San Diego, La Jolla, California, 92093, USA. [3] Department of NanoEngineering, University of California, San Diego, La Jolla, California 92093, USA. Correspondence and requests for materials should be addressed to Y.F. (email: fainman@ece.ucsd.edu).

The response of materials to electromagnetic fields is often described by the relative electrical permittivity, $\varepsilon$, and magnetic permeability, $\mu$ (ref. 1). In general, $\varepsilon$ and $\mu$ are complex-valued, frequency- and wave-vector-dependent, tensorial quantities, the elements of which relate the spatial directions of the driving and responding fields[2]. In some naturally existing[3] and artificially engineered[4] materials, diagonal elements of the $\varepsilon$ or $\mu$ tensor have opposing sign in a particular range of frequencies. For these materials, the relationship between the wave vector and frequency describes a hyperboloid (hyperbola) of indefinite volume (area) in three-dimensional (two-dimensional, 2D) reciprocal space[5]. Consequently, such materials have generally been called indefinite media and their artificially engineered versions have been dubbed hyperbolic metamaterials (HMMs), which exhibit a diverging density of states in the lossless effective medium limit[6].

Physical realizations of indefinite media and HMMs include magnetized plasmas at radio frequencies[7], arrays of ceramic particles at microwave frequencies[8], nanostructured graphene at far-infrared and THz frequencies[9], 2D van der Waals crystals[10,11] and highly doped semiconductors at mid-infrared frequencies[12], doped metal oxide-based multilayers at near-infrared frequencies[13,14], and noble metal-based multilayers[15–18] and nanowire arrays[19–21] at visible frequencies. In all realizations, the material response is highly anisotropic, with the extreme case being simultaneous metallic and dielectric behaviour for fields of orthogonal polarization states.

At infrared and visible frequencies, coupling between surface plasmon polaritons at adjacent metal-dielectric interfaces leads to the formation of so-called volume[22] or bulk[21,23] plasmon polaritons (BPP), which are waves that propagate normal to the interfaces. Features of BPPs include deeply subwavelength confinement of energy, extremely short propagation lengths and extremely short lifetimes[23]. Consequently, BPPs find applications in sub-diffraction-limited imaging[24], field-enhanced nonlinear devices[25,26], super-Planckian heat transfer[27,28], perfect absorbers (Riley, C.T. et al. manuscript submitted), spontaneous emission engineering[16–18,29,30] and ultrafast switching[20]. In addition, the large momentum of BPPs has found use in asymmetric transmission devices[31] and in HMM-clad waveguides with highly tailorable modal dispersion[32–35].

Two fundamental challenges associated with HMMs include large rates of dissipation and large impedance mismatch with their environment. Both of these may be considered forms of wasted energy for fields coupling to BPPs from an external source. Consequently, the exciting scientific phenomena and technological applications offered by indefinite media and HMMs tend to be challenging to observe and extremely inefficient.

To improve the efficiency of HMMs, strategies include (1) mitigation of internal loss and (2) mitigation of insertion loss. Use of high-quality materials and fabrication processes can help mitigate both internal and insertion losses. For example, single-crystal silver (Ag) was grown epitaxially in a hyperbolic metasurface (HMS) geometry, enabling long propagation lengths and the observation of the plasmonic spin Hall effect at visible frequencies[36]. The placement of quantum emitters with respect to the HMM is also an important factor related to both internal and insertion losses. Theoretical[37] and experimental[18] work has shown that the emission rate and intensity of emitters is significantly higher when the emitters are located within the HMM rather than adjacent to it. To this end, an HMM was recently fabricated with a quantum well layer in a rolled-up geometry[38]. Although meeting the design guideline of placing emitters within the HMM, this geometry suffers from extremely high insertion loss, as the BPPs cannot be efficiently excited due to impedance mismatch with the environment.

Grating couplers have been shown to significantly reduce insertion loss, using a variety of geometries and fabrication processes. For example, one-dimensional rectilinear gratings[16] and so-called hyper-crystal[29] gratings made by focused ion beam were used to show emission enhancement of organic dyes and 2D transition metal dichalcogenides, respectively. In addition, nanohole array gratings[17] and bullseye[18] gratings made by electron-beam lithography (EBL) and dry etching were used with dyes and quantum dots, respectively. Although dramatically improving coupling between BPPs and their environment, use of these gratings necessarily increases fabrication complexity.

In addition to mitigation of internal and insertion losses, theoretical studies have shown that losses in HMM may be compensated with gain[39–45]. These studies generally assume replacement of the passive constituent dielectric with an active gain material and have yielded encouraging results that include the possibility of signal amplification under particular conditions. Namely, the signal frequency must be sufficiently far from the plasma frequency of the constituent metal[44,45], whereas signal propagation should occur in the direction normal to the metal-dielectric interfaces[40,42,44]. Gain media considered have included organic dye molecules[40–43], which are relatively easy to blanket deposit atop multilayer or nanowire array HMMs. However, dyes generally suffer from a high propensity to photo-bleach, poor integration with electronics and low quantum yield at near-infrared frequencies. Inorganic semiconductors have also been considered as gain media[44,45], the fabrication of which into multilayer structures with noble metals faces technical challenges, because semiconductor growth processes generally do not permit alternating deposition with metals. Nonetheless, good photo-stability, the potential for electrical gating or carrier injection using appropriately doped heterostructures and their high quantum efficiency makes III–V compounds an attractive gain media for HMMs operating at visible and near-infrared frequencies. To date, however, no physical realization of HMMs has successfully placed emitters or gain media fully throughout the structure, while maintaining low insertion loss and accessibility to the environment without the need of additional grating couplers.

In this work, we demonstrate luminescent HMS (LuHMS) that address the fundamental challenges associated with HMMs. Namely, we design, fabricate and characterize HMS in which semiconducting multiple quantum wells (MQWs) simultaneously function as the emission source and constituent dielectric in a multilayer architecture combined with Ag, with optical axis in the plane of the substrate. Consequently, optimal light–matter interaction and efficient extraction of in-plane surface modes[46] are simultaneously achieved, to counter both internal and insertion losses. Furthermore, the unique multilayer realization enables characterization of hyperbolic dispersion by the extreme polarization anisotropy (PA) of absorption and emission, achieved experimentally through normally incident micro-photoluminescence (PL) measurements. The demonstrated LuHMS are inherently amenable to electronic integration and represent a significant advancement in active plasmonics, where applications enabled by deeply subwavelength energy confinement may be improved through the judicious choice of constituent materials, geometry and fabrication processes.

## Results

**Subwavelength multilayer metal-semiconductor nanostructures.** Our extremely anisotropic LuHMS consists of alternating layers of Ag and indium gallium arsenide phosphide (InGaAsP) MQW, shown in Fig. 1. InGaAsP MQW pillars of 100 nm height and 40 nm width, separated by 40 nm-wide trenches, are defined by EBL and reactive ion etching, as shown in Fig. 1a (see Methods). After etching, Ag is blanket deposited by sputtering, partially

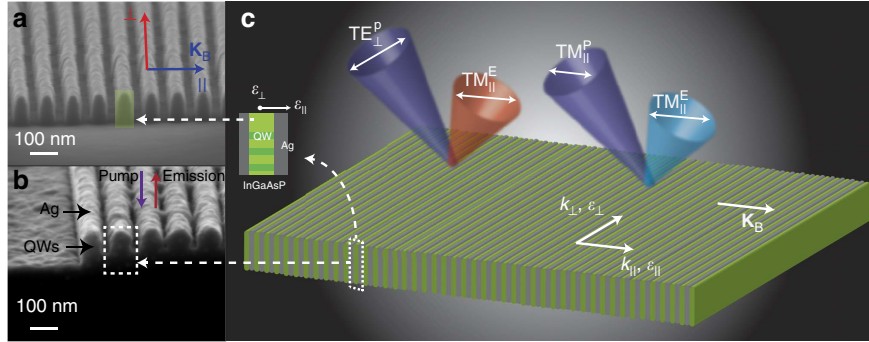

**Figure 1 | LuHMS based on nanostructured Ag/InGaAsP MQW.** (**a**) InGaAsP MQW pillars of 100 nm height and 40 nm width, separated by 40 nm trenches, are defined by EBL and reactive ion etching. (**b**) Ag is deposited by sputtering, partially filling the trenches to create a multilayer LuHMS with 80 nm period. (**c**) Optical pumping($TM_{||}^{P}\ or\ TE_{\perp}^{P}$)of the LuHMS results in collected emission polarized predominantly parallel ($TM_{||}^{E}$) to the metacrystal Bloch vector, $\mathbf{K_B}$. The wavelength of peak emission and PL intensity depend strongly on pump polarization. Emission from the LuHMS blue shifts as the pump polarization changes from $TE_{\perp}^{P}$ to $TM_{||}^{P}$ due to increasing pump absorption. In addition, PL spectra of the LuHMS differ significantly from that of control MQW, regardless of pump polarization, due to a wavelength and pump power dependence of the direction of energy propagation on the surface.

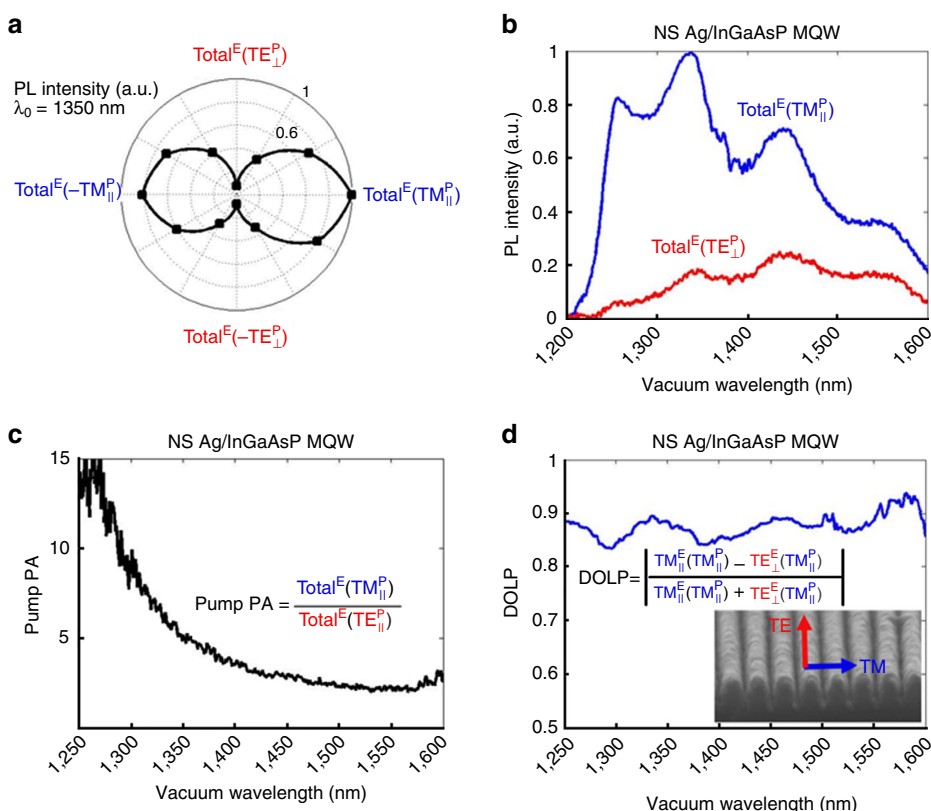

**Figure 2 | Demonstration of extreme PA in LuHMS.** (**a**) Dependence of total emission ($Total^E$) on pump polarization at emission wavelength of 1,350 nm. Maxima and minima are clearly observed for $TM_{||}^{P}$ and $TE_{\perp}^{P}$ pumping, respectively (**b**) Total PL spectra of the LuHMS for parallel and normal polarized pump. The shape of the spectra differ due to band-filling effects. (**c**) Pump PA of total emission calculated from **b**. The pump PA increases with frequency due to band-filling associated with the more efficiently absorbed pump polarization. (**d**) DOLP of emission for parallel-polarized pump. The DOLP is close to unity, indicating nearly linearly polarized light in the direction parallel to $\mathbf{K_B}$. Inset to **d** shows SEM of nanostructure and definition of polarizations.

filling the trenches to form alternating nanostructured (NS) Ag/InGaAsP MQW layers, shown in Fig. 1b. The resulting multilayer has a period of $\Lambda_B \approx 80$ nm, 15–20 times smaller than the vacuum emission wavelength of the constituent MQW, and is therefore suitable for description by the effective medium approximation (EMA)[47].

Using the EMA, we describe the linear optical response of the LuHMS with the diagonal effective permittivity tensor, $\boldsymbol{\varepsilon} = [\varepsilon_\perp, 0, 0; 0, \varepsilon_\perp, 0; 0, 0, \varepsilon_{||}]$, with tensor elements defined with

respect to the metacrystal Bloch vector, $\mathbf{K_B}$ (see Fig. 1). In the wavelength range of interest, $1,000\,\text{nm} < \lambda_0 < 1,600\,\text{nm}$, the tensor elements have the following properties in the absence of external pumping (see Supplementary Figs 1 and 2, and Supplementary Note 1),

$$\varepsilon_\perp = \varepsilon'_\perp + i\varepsilon''_\perp, \ \varepsilon'_\perp < 0, \ \varepsilon''_\perp > 0 \tag{1}$$

$$\varepsilon_{||} = \varepsilon'_{||} + i\varepsilon''_{||}, \ \varepsilon'_{||} > 0, \varepsilon''_{||} > 0 \tag{2}$$

 

In Equations (1) and (2), the single and double primes denote the real and imaginary parts, respectively, and the time-convention of $\exp(-i\omega t)$ has been chosen such that positive $\varepsilon''$ indicates dissipation. Intuitively, Equations (1) and (2) state that the LuHMS resembles InGaAsP MQW, a strongly absorbing semi-conductor, for waves polarized parallel to $\mathbf{K}_B$, whereas it resembles Ag, a strongly scattering metal, for waves polarized normal to $\mathbf{K}_B$ (Supplementary Note 1). Under the influence of optical pumping, $\varepsilon''_{\perp}$ and $\varepsilon''_{\parallel}$ change dramatically, with $\varepsilon''_{\parallel}$ becoming negative for sufficiently large pump powers[44]. Optical pumping of the LuHMS results in emission polarized predominantly parallel to $\mathbf{K}_B$, shown schematically in Fig. 1c, regardless of pump polarization. Throughout this work we use the notation of transverse magnetic ($TM^{\alpha}_{\parallel}$) and transverse electric ($TE^{\alpha}_{\perp}$) for waves polarized parallel and normal to $\mathbf{K}_B$, respectively, with superscript $\alpha = P$ or $\alpha = E$ for pump and emission, respectively.

As the pump polarization changes from $TE^P_{\perp}$ to $TM^P_{\parallel}$, the wavelength of peak emission blue shifts and the integrated PL intensity increases due to a larger inversion density induced by parallel pumping. It is noteworthy that because the constituent materials of our system are non-magnetic, $\boldsymbol{\mu} = [1,0,0;0,1,0;0,0,1]$.

**Extreme PA of PL.** We verify hyperbolic dispersion of the fabricated LuHMS experimentally by extreme PA of PL. A linearly polarized, pulsed Nd:YAG laser with vacuum wavelength of $\lambda_0 = 1,064$ nm was used to photo-excite carriers in the MQW at room temperature (see Methods, Supplementary Fig. 3 and Supplementary Note 2). Unless stated otherwise, average power, average intensity and peak intensity of the pump were 10 mW, 9.9 kW cm$^{-2}$ and 2.7 MW cm$^{-2}$, respectively. Under $TM^P_{\parallel}$ ($TE^P_{\perp}$) pumping, the total PL signal reaches a maximum

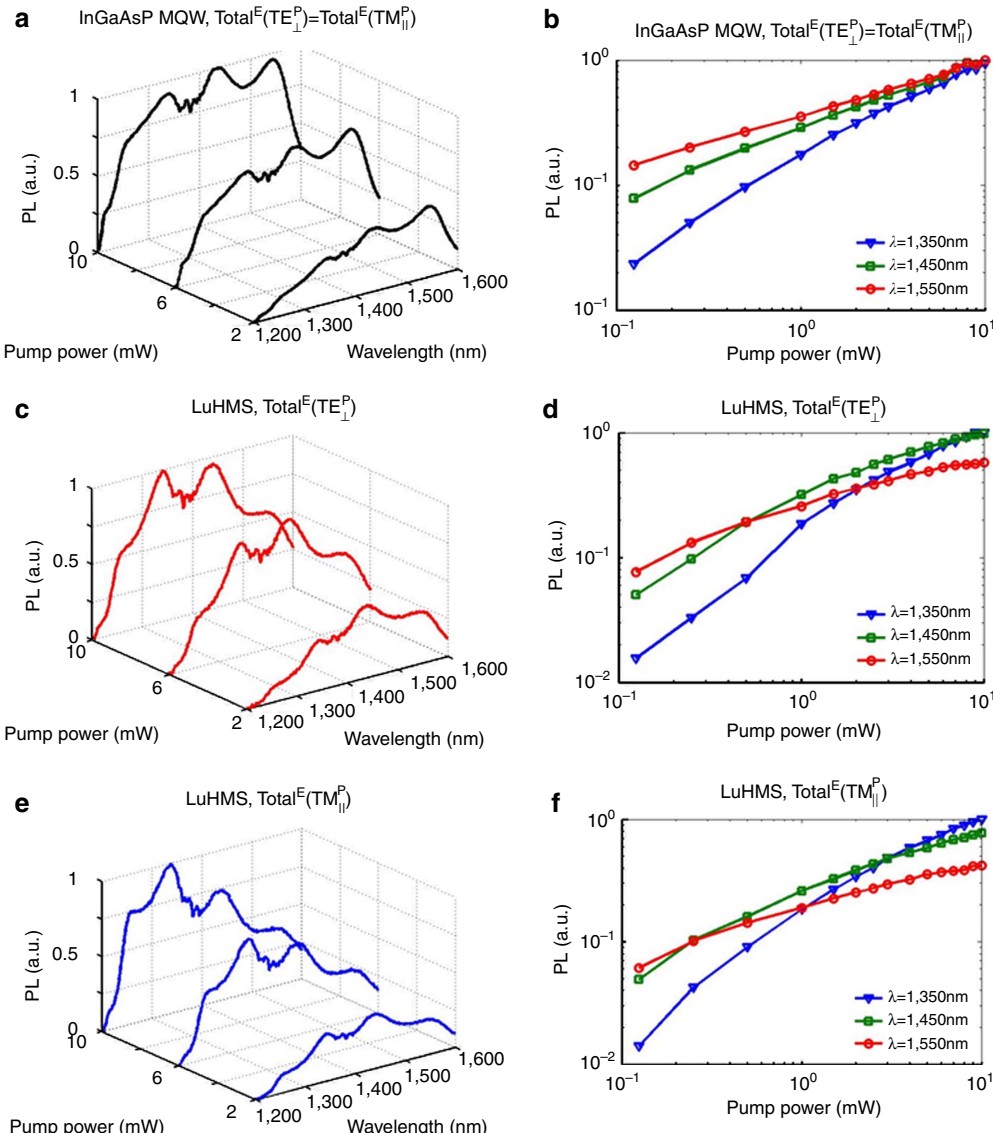

**Figure 3 | Demonstration of modified emission spectra in LuHMS.** (**a,c,e**) Evolution of total emission (Total$^E$) with pump power on linear scale and (**b,d,f**) log–log plot of total emission as a function of pump power for select wavelengths. (**a,b**) Control InGaAsP MQW. (**c,d**) LuHMS excited by pump polarized normal and (**e,f**) parallel to metacrystal Bloch vector, $\mathbf{K}_B$. Significant blue shifting of peak PL occurs in the LuHMS relative to the control MQW, which is attributed to the wavelength dependence of the principal direction of energy propagation in the LuHMS (see Supplementary Fig. 9 and Supplementary Note 4). Additional blue shifting of peak PL occurs in LuHMS under parallel-polarized pumping due to greater pump absorption. The simultaneous dependence of PL on pump polarization and electronic and optical densities of states suggests an avenue for engineering tunable 'meta-gain' materials (Supplementary Fig. 14 and Supplementary Note 8).

(minimum), shown for the emission wavelength of $\lambda_0 = 1,350$ nm in Fig. 2a and the entire spectrum in Fig. 2b. The PL spectrum broadens and the wavelength of peak PL blue shifts as the pump polarization evolves from $TE_\perp^P$ to $TM_\parallel^P$. As the pump polarization becomes more aligned with $\mathbf{K}_B$, absorption increases, exciting carriers to higher energy states in the MQW. Numerical simulations confirm the observed extreme absorption anisotropy and the efficacy of the EMA (see Supplementary Fig. 4 and Supplementary Note 3). Figure 2c quantifies the pump PA, as the ratio of total PL generated by the parallel and normal polarized pumps. The nonlinear dependence of PA on wavelength reflects a combinatorial effect of directional propagation associated with enhanced optical density of states and filling of the electronic density of states according to the Pauli exclusion principle (see Supplementary Note 4).

In addition to the strong dependence of total PL on pump polarization, the emission itself is highly polarized. The degree of linear polarization (DOLP) is defined as the difference between orthogonal polarization components of the emission relative to the total emission, that is, $DOLP = |TM_\parallel^E - TE_\perp^E|/(TM_\parallel^E + TE_\perp^E)$, and is shown in Fig. 2d for parallel-polarized pumping. Regardless of pump polarization, the emission is predominantly polarized parallel to $\mathbf{K}_B$. This occurs despite the fact that emission of bare InGaAsP MQW is predominantly polarized normal to $\mathbf{K}_B$ (see Supplementary Fig. 5 and Supplementary Note 2). The highly polarized emission of the LuHMS may be attributed to the fact that, in

principle, hyperbolic media only support modes with an electric field component parallel to $\mathbf{K}_B$ (ref. 23). Analytical calculations confirm that the LuHMS supports only TM-polarized modes (Supplementary Fig. 6 and Supplementary Note 3) and numerical simulations illustrate polarization filtering of the emission from the MQW, with $TM_\parallel^E$ transmitted to the far-field and $TE_\perp^E$ reflected towards the substrate (Supplementary Fig. 4). Consequently, only parallel-polarized emission from the MQW couples to the structure and reaches the far-field continuum by phase matching (see Supplementary Fig. 7 and Supplementary Note 5).

**Blue-shifting of PL spectra despite less absorption.** Comparison of emission from the LuHMS with that from the control MQW sheds additional light on the interplay of electronic and optical density of states with the pumping condition. For all pump powers shown in Fig. 3a,b, the transition at ~1,550 nm, between the first conduction and heavy-hole subbands, dominates the spectrum of the control MQW (see Supplementary Fig. 8 and Supplementary Note 2). For identical pump powers, the peak emission from the LuHMS is blue shifted relative to the control MQW. This occurs despite the fact that the blanket-sputtered Ag inevitably scatters the pump before it reaches the MQW. Under normal-polarized pumping, shown in Fig. 3c,d, the transition at ~1,450 nm, between the first conduction and light-hole subbands, dominates the spectrum of LuHMS at high power.

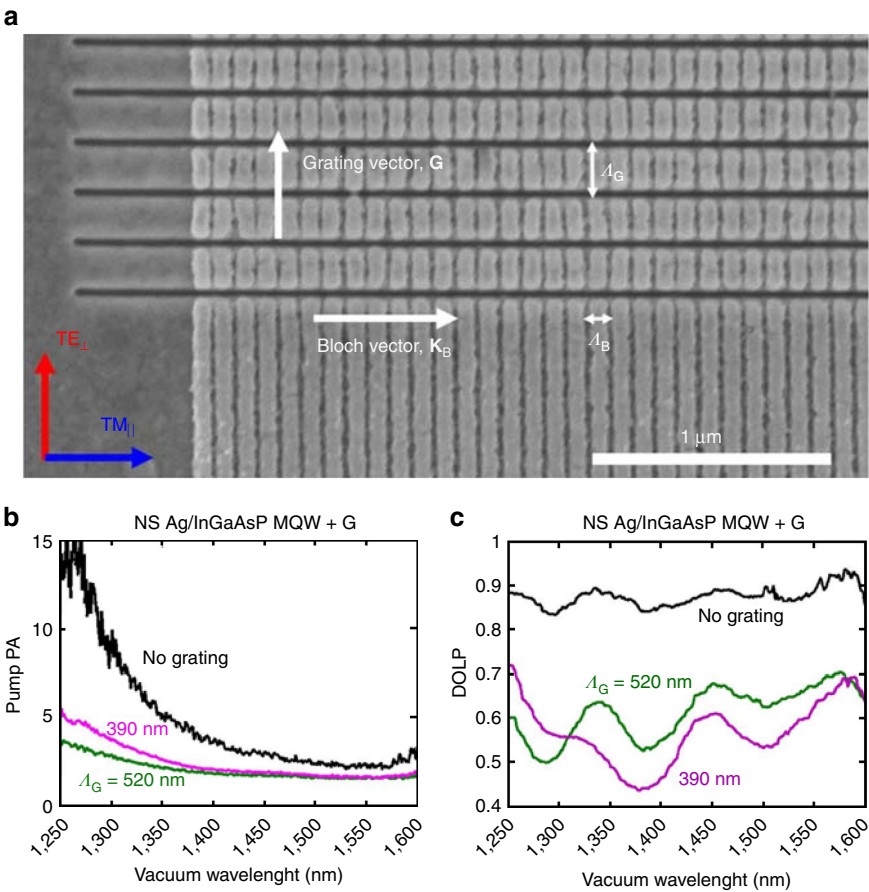

**Figure 4 | Demonstration of PA reduction via grating coupling. (a)** LuHMS with wavelength-scale grating coupler fabricated with focused ion beam milling. The grating is designed to improve, both in-coupling of the pump and out-coupling of emission polarized normal to the metacrystal Bloch vector, $\mathbf{K}_B$. Comparison of (**b**) Pump PA and (**c**) DOLP in the absence and presence of gratings. Improved coupling of normal-polarized light reduces the pump PA and DOLP, consistent with behaviour of simulated LuHMS. The reduction in anisotropy occurs across all measured grating periods and is robust to small changes in geometric parameters of the LuHMS (see Supplementary Fig. 10 and Supplementary Note 6).

Conversely, under parallel pumping, shown in Fig. 3e,f, a transition at ~1,350 nm between the second conduction and heavy-hole subbands dominates the high-power spectrum.

Greater absorption may explain the blue shift of the parallel-pumped LuHMS relative to the normal-pumped LuHMS; however, it cannot explain blue shifts of LuHMS emission relative to the control MQW. Although the presence of Ag increases scattering, it also modifies the optical density of states, effectively filtering different spectral components of the emission. This may be understood through the wavelength dependence of the direction of energy propagation in the LuHMS. (see Supplementary Fig. 9 and Supplementary Note 4). The resonance cone angle, which defines the principal direction of energy propagation relative to $\mathbf{K}_B$, increases with wavelength. Consequently, shorter (longer) wavelengths of emission experience less (more) attenuation as they propagate along the LuHMS. Furthermore, the resonance cone angle depends on pump intensity, suggesting that peak emission wavelengths may be tuned by the directional properties of the LuHMS, in addition to electronic band filling.

**Reduction of PA and PL enhancement**. To further verify hyperbolic dispersion in the NS Ag/InGaAsP MQW system, we subsequently designed and fabricated LuHMS with wavelength-scale grating couplers, shown in Fig. 4a. We designed the grating couplers to increase absorption and emission of light polarized normal to $\mathbf{K}_B$ and thereby reduce the pump PA and DOLP (see Methods and Supplementary Fig. 6). Owing to the unique multilayer design, parallel polarized emission does not require a grating (Supplementary Figs 6 and 7, and Supplementary Note 5). PL measurements show a significant reduction in PA for LuHMS with grating periods of $\Lambda_G = 390$ nm and $\Lambda_G = 520$ nm (Fig. 4b). Regardless of grating period and deviations from fabrication specifications, the PA is always reduced (see Supplementary Fig. 10 and Supplementary Note 6). In addition, emission from the LuHMS with gratings is less polarized, as shown by a reduction in DOLP in Fig. 4c. Grating coupling therefore provides a control mechanism for pump PA and emission DOLP.

Beyond controlled extreme PA, the LuHMS also exhibits intensity enhancement of PL (see Supplementary Figs 11 and 12, and Supplementary Note 7). Using a reverse excitation technique where the pump excites the active material from the substrate, we observed PL enhanced by ~350% and ~25% in the LuHMS, relative to the control InGaAsP MQW and flat Ag/InGaAsP MQW interface, respectively. The observed intensity enhancement occurs across the entire emission spectrum and range of pump powers investigated.

## Discussion

To better understand the absorption and emission behaviour of our fabricated samples, we performed numerical finite-difference time-domain simulations at the pump and emission wavelengths, modelling the LuHMS as both an exact nanostructure and according to the EMA (see Methods, Supplementary Fig. 4 and Supplementary Note 3). The qualitative agreement between simulated results and experimental measurements confirms that the fabricated samples exhibit hyperbolic dispersion. Close quantitative agreement between results for the exact nanostructure and EMA additionally shows the utility of the EMA in designing the LuHMS. To dispel the counter argument that PA occurs equally on all scales of periodicity, we also simulated pump absorption over a range of periods for which the EMA loses its validity (Supplementary Fig. 6 and Supplementary Note 3). The PA weakens significantly with increasing period, confirming that hyperbolic dispersion is necessary to observe extreme anisotropy of PL.

Finally, to confirm that Ag was needed in addition to nanostructuring for the observation of pump PA, we created additional samples and measured the pump polarization-dependent PL of etched InGaAsP MQW before and after Ag deposition (Supplementary Fig. 13 and Supplementary Note 2). The lack of pump PA before Ag deposition and dramatic dependence of PL intensity and lineshape on polarization after deposition unambiguously confirms that Ag, and therefore hyperbolic dispersion, is indeed necessary to achieve extreme PA.

We have demonstrated LuHMS using extreme PA of PL. The demonstrated dependence of LuHMS behaviour on pump power and polarization suggest new opportunities for engineering 'meta-gain' media, enabled by deeply subwavelength nanostructuring of luminescent materials (Supplementary Fig. 14 and Supplementary Note 8). The inherent tunability enables enhanced functionality of hyperbolic dispersion, otherwise limited by excessive dissipation[40], and may prove useful for amplification and lasing of plasmonic modes[48]. Advantages of the demonstrated LuHMS, compared with prior state-of-the-art, include distribution of emitters throughout the entire HMM, coupling of high-$k$ states to vacuum states without the need of a grating, PA of absorption and emission at normal incidence and potential for electronic addressability. Avenues for future research in LuHMS include studying the temporal dynamics of emission, electronic transport and field-enhanced nonlinear properties, with device applications in optical sensing, computing and communications.

## Methods

**Nanofabrication of deeply subwavelength multilayer structures.** The heterostructures were grown offsite (OEpic) and consist of 300 nm undoped InGaAsP MQW epitaxially grown on an InP substrate. The MQW consists of nine 10 nm wells ($In_{0.564}Ga_{0.436}As_{0.933}P_{0.067}$) separated by 20 nm barriers ($In_{0.737}Ga_{0.263}As_{0.569}P_{0.431}$), with the exception that the final barrier is 30 nm. A 10 nm InP capping layer over this last barrier terminates the growth. Processing began with wet-etching of the InP capping layer with a 3:1 solution of hydrochloric acid to deionized water, followed by spin-coating of ~40 nm of hydrogen silsesquioxane (HSQ) EBL resist onto pristine InGaAsP cut into a 1 cm × 1 cm sample area. The pattern was written using EBL (Vistec EBPG 5200) with doses in the range of 850–950 $\mu C\,cm^{-2}$ at 30 kV and beam current of 3 nA. After exposure, the HSQ is developed and behaves as a glass-like mask for the etching of InGaAsP pillars. The pillars were defined by reactive ion etching (Trion) using gas flow rates of 10 sccm $CH_4$, 40 sccm $H_2$ and 20 sccm Ar at a chamber pressure of 30 mTorr, temperature of 35 °C and RF power of 100 W. Subsequently, the HSQ was removed by a 10 s dip in a solution of 49% hydrofluoric acid. The Ag film was grown at room temperature under high vacuum ($3 \times 10^{-9}$ Torr) by DC magnetron sputtering. To fill the trenches uniformly, the Ag target was positioned in a sputter gun directly below the etched InGaAsP and the sample holder was continuously rotated during the deposition. Once growth was initiated, the Ag film was deposited at a pressure of 1.3 mTorr with Ar gas flow of 50 sccm and DC power of 20 W. For some samples, an additional focused ion beam (FEI Scios) step followed. Gratings of ~50–100 lines, with line width of 30 nm and variable pitch (period), were defined orthogonally to the metacrystal Bloch vector, $\mathbf{K}_B$, with a Ga ion beam of 30 kV voltage and 1.5 pA current.

**Mircophotoluminescence characterization.** A pulsed Nd:YAG pump laser (SPI G4), emitting 12 ns pulses at a repetition rate of 300 kHz and freespace wavelength of $\lambda_0 = 1,064$ nm was used to optically excite the sample at normal incidence. With the aid of an infrared imaging system (Indigo/FLIR Alpha) with a 50 μm × 50 μm field-of-view, the pump was focused to a ~8 μm spot size, exciting the LuHMS using a × 20 magnification microscope objective with 0.4 numerical aperture (Nikon). With a maximum average power of 15 mW, the maximum average and peak intensities of the Gaussian beam used were 14.9 $kW\,cm^{-2}$ and 4.1 $MW\,cm^{-2}$, respectively. Before reaching the sample, the pump was passed through a polarizing beam splitter and polarization rotator, providing complete control of the direction of linear polarization (see Supplementary Fig. 3 and Supplementary Note 2). Broadband PL was collected with the same microscope objective, passed through a linear polarizer and directed to a monochromator with 1 mm slit width and 2 nm spectral resolution (DK480 1/2 m). An InGaAs photodiode (EOSystems IGA-010-TE2-H), cooled thermoelectrically to −30 °C, received the signal from the monochromator. The pump beam was chopped at a frequency of 1,000 Hz and synchronized with the detection system using a digital lock-in amplifier (Stanford Research Systems). All characterization took place at room temperature.

 

**Finite-difference time-domain simulations.** PA of the pump and emission were simulated using a 2D finite-difference time-domain model (Lumerical, see Supplementary Figs 4 and 6, and Supplementary Note 3). Periodic boundary conditions and perfectly matched layers were used in the direction parallel and normal to the metacrystal axis. For pump simulations, a quasi-monochromatic plane wave source was launched at normal incidence from air onto the structure. For emission simulations, a plane wave source was launched at normal incidence from the substrate towards the structure. The simulated structures consisted of both InGaAsP pillars clad with Ag and the EMA. The complex dielectric functions of silver and InGaAsP were taken from a combination of experimental data[49,50] and theoretical models[51,52]. Mesh size for all simulations was 3 nm × 3 nm.

**Data availability.** The data that support these findings are contained in the manuscript and the Supporting Information.

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

## Acknowledgements

We thank Dr Maribel Montero for EBL services and Babak Bahari for helpful discussions. This work was supported by the Office of Naval Research Multidisciplinary Research Initiative (N00014-13-1-0678), the National Science Foundation (NSF) (ECE3972 and ECCS-1229677), the NSF Center for Integrated Access Networks (EEC-0812072, Sub 502629), the Defense Advanced Research Projects Agency (N66001-12-1-4205) and the Cymer Corporation.

 7

## Author contributions

J.S.T.S., F.V. and Y.F. conceived the project and designed the experiment. F.V., J.S.T.S., S.A.M., L.F. and C.T.R. fabricated the samples. J.S.T.S. measured the samples, performed data analysis and analytical calculations. F.V., J.S.T.S. and S.S. performed numerical simulations. B.K., E.E.F., Z.L. and Y.F. provided consultation. J.S.T.S. wrote the manuscript and all authors partook in editing and reviewing the manuscript.

## Additional information

**Competing financial interests**: The authors declare no competing financial interests.

**How to cite this article**: Smalley, J. S. T. *et al.* Luminescent hyperbolic metasurfaces. *Nat. Commun.* **8**, 13793 doi: 10.1038/ncomms13793 (2017).

**Publisher's note**: 

