## [Peer Review File · Nature Communications]

Reviewers' comments:

Reviewer #1 (Remarks to the Author):

This is an interesting paper that is clearly written with good illustrations which describes a manufactured material that responds according to the polarisation of incident electromagnetic radiation. I have a few suggestions for improvement.

1. Lines 10-12. Would it be possible for the authors to provide actual examples of devices where "hyper-imaging, enhanced and directional emission, ultrafast modulation and efficient higher harmonic generation" could be used? Is hyper-imaging related to hyper spectral imaging? I think that a bit more information here would enable readers to gain the significance more readily. Since this work was done with infrared is it just applicable to infrared imaging?
2. Lines 16-17 "... function as light-emitters and the constituent dielectric...". Maybe the authors could expand on the significance of this feature of this material. Does this enable large scale integration of chips with this device, or does it make LSI more efficient? Does the combined light emitting/dielectric properties make the layer thinner enabling heat to be removed more efficiently?
3. Figure 1. It might be useful to add arrows indicating the direction of the incident and reflected light. What's the control MQW? Non InGaAsP pillars? In part (b) of the diagram is the undulation 'floor' to the left of the QW pillars the Ag and indicate the level of the Ag in the trenches between the QWs? I think it would be useful to indicate whether this is the case.
4. Figure 2. What are the band-filling effects?
5. Figure 3 (b), (d), (f). I think that trend lines would show the blue shifting more clearly.

Reviewer #2 (Remarks to the Author):

This manuscript reports the use of a silver coated subwavelength grating: a.k.a hyperbolic metasurface to modify the spontaneous emission from InGaAsP quantum wells. The semiconducting material serves the purpose of both the dielectric and the emitter resulting in the luminescent hyperbolic metasurface (LuHMS).

The manuscript DOES NOT provide any convincing evidence for the enhancement in spontaneous emission due to the increased density of states in the LuHMS. Some of the co-authors have demonstrated this effect much more convincingly in several previous papers. All that is shown in the present work is the modification in the emission pattern and polarization anisotropy. They compare the results to that of plain QW. However the question is how does the emission pattern look when just a dielectric sub-wavelength grating (minus Ag) is used?

However this still does not address the main concern - what was gained by etching through quantum wells and putting metal right next to it? Is polarization anisotropy at the expense of quantum yield the biggest gain? It appears that all that is happening is polarization filtering -can't that be done with pure dielectric structures? How bad did the quantum yield of the quantum well become?

In fact the same filtering effect is used to explain the blue shift in the emission.

The authors also carry out polarization anisotropy measurements on wavelength scale gratings and show that it is decreased when compared to LuHMS. They attribute this to the hyperbolic dispersion. However here again it is not clear what Ag does - will just a sub-wavelength grating do the job?

Simulations using effective medium approximation has been used to explain the observed effects.

In summary I find the results to be of incremental nature, not convincing of the claims and most importantly limited in scope and of likely interest to a small sub-section of researchers working on hyperbolic metamaterials. The only key result is the polarization filtering and it is not even clear if one needs to go through all this hard fabrication to achieve the same goal!

I recommend rejecting this manuscript.

Reviewer #3 (Remarks to the Author):

The manuscript presents an interesting new development in the field of active metamaterials - that combines the density of states super-singularity of the hyperbolic media with the simplicity of the fabrication offered by the metasurfaces. In future, this approach may eventually lead to a practical light-emitting device, as well as a new platform to study novel elements of light-matter interaction in artificial media. I therefore believe that the manuscript satisfies the stringent criteria of Nature Communications, and should be published.

However, as the two key elements in the proposed approach are (i) the super-singularity of the density of states in hyperbolic media, and (ii) the surface states with hyperbolic dispersion, the authors should reference the original papers that introduce these two concepts. To the best knowledge of the Referee, the photonic density of states super-singularity in hyperbolic media was first introduced in

I. I. Smolyaninov and E. E. Narimanov. Phys. Rev. Lett. 105, 067402 (2010),

while the first discussion of surface states with (in-plane) hyperbolic dispersion was presented in

Z Jacob and E.E. Narimanov, Applied Physics Letters 93 (22), 221109 (2008),

and the authors must reference these two papers (unless they find earlier work on the subject) in the introduction of the revised version of their manuscript.

Response to reviewers' comments for "Luminescent Hyperbolic Metasurfaces" (NCOMMS-16-09010A)

Reviewer #1 (Remarks to the Author):

This is an interesting paper that is clearly written with good illustrations which describes a manufactured material that responds according to the polarisation of incident electromagnetic radiation. I have a few suggestions for improvement.

1. Lines 10-12. Would it be possible for the authors to provide actual examples of devices where "hyper-imaging, enhanced and directional emission, ultrafast modulation and efficient higher harmonic generation" could be used?

Indeed, we had provided concrete references when writing about these applications of hyperbolic dispersion, including [1] for hyper-imaging, [2–4] for enhanced and directional emission, [5] for ultrafast modulation, and [6–8] for efficient higher harmonic generation. Due to the original publication limit of 30 references, however, we decided not to include additional references, as we felt the references that we did include were more vital for the reader. In the revised manuscript we have added the application of super-Planckian heat transfer [9].

Is hyper-imaging related to hyper spectral imaging? I think that a bit more information here would enable readers to gain the significance more readily. Since this work was done with infrared is it just applicable to infrared imaging?

Hyper-imaging refers to imaging below the diffraction limit using the structures with hyperbolic dispersion. In principle, hyper-imaging can occur at any wavelength range wherein materials exhibit hyperbolic dispersion. Hyper-spectral imaging, on the other hand, refers to imaging of a two-dimensional scene across a very broad spectral range. Therefore, hyper-imaging and hyper spectral imaging are not directly related.

2. Lines 16-17 "... function as light-emitters and the constituent dielectric...". Maybe the authors could expand on the significance of this feature of this material. Does this enable large scale integration of chips with this device, or does it make LSI more efficient? Does the combined light emitting/dielectric properties make the layer thinner enabling heat to be removed more efficiently?

As discussed in [10], the spontaneous emission enhancement—Purcell factor and radiative enhancement—can be maximized when the light emitters are placed within the hyperbolic metamaterial (HMM). We have designed the LuHMS with this in mind. Emitters are distributed throughout the resulting HMM because the emitters and constituent dielectric are the same material. Conversely, other reported HMMs have incorporated light emitters in less systematic ways. For example, in [2] and [3] dye molecules were placed next to, but not within the HMM structure. In [4] a quantum dot was placed in a single location within the HMM, but not throughout the HMM as in our work. To more clearly explain the advantages of the LuHMS we have added reference [10] to the revised manuscript.

3. Figure 1. It might be useful to add arrows indicating the direction of the incident and reflected light. What's the control MQW? Non InGaAsP pillars? In part (b) of the diagram is the undulation 'floor' to the

left of the QW pillars the Ag and indicate the level of the Ag in the trenches between the QWs? I think it would be useful to indicate whether this is the case.

We agree with the reviewer and have modified Fig. 1 to indicate the direction of incident and reflected light in Fig. 1 (b). We have also indicated where the Ag and QWs in the same figure. Regarding the control MQW, we characterized MQW samples with and without etched nanostructures. Both samples exhibit negligible pump polarization dependence and have negligible difference between them. Therefore, for the sake of our experiments, the InGaAsP MQW with and without etching are equivalent, both functioning as the control MQW. We have added a sentence in the manuscript for clarification.

4. Figure 2. What are the band-filling effects?

Band-filling effects refer to the blue-shifting of emission due to the increasing separation of quasi-Fermi levels in the InGaAsP MQW, with increasing absorption of the pump beam. In the revised manuscript, we have added a sentence to clarify our meaning of this phrase.

5. Figure 3 (b), (d), (f). I think that trend lines would show the blue shifting more clearly.

In the revised manuscript we have added trend lines to Fig. 3 (b,d,f).

We thank Reviewer 1 for his/her comments and suggestions.

Reviewer #2 (Remarks to the Author):

This manuscript reports the use of a silver coated subwavelength grating: a.k.a hyperbolic metasurface to modify the spontaneous emission from InGaAsP quantum wells. The semiconducting material serves the purpose of both the dielectric and the emitter resulting in the luminescent hyperbolic metasurface (LuHMS).

The manuscript DOES NOT provide any convincing evidence for the enhancement in spontaneous emission due to the increased density of states in the LuHMS.

The main idea of the submitted manuscript is the experimental demonstration of hyperbolic dispersion in the Ag/InGaAsP MQW system. While other authors have demonstrated hyperbolic dispersion through the enhancement of the spontaneous emission rate and/or intensity, we chose to demonstrate hyperbolic dispersion by extreme polarization anisotropy of photoluminescence. In fact, to the best of our knowledge, we are the first to characterize hyperbolic dispersion using this technique. In the originally submitted manuscript, we made no claim of enhanced spontaneous emission, neither rate nor intensity.

Some of the co-authors have demonstrated this effect much more convincingly in several previous papers. All that is shown in the present work is the modification in the emission pattern and polarization anisotropy.

We have shown that the emission from the metamaterial depends strongly on polarization, whereas the control MQW does not. Our use of pump polarization anisotropy and degree-of-linear-polarization of the emission is, to the best of our knowledge, an original means of characterizing hyperbolic dispersion. Therefore, using this technique rather than enhanced spontaneous emission is a matter of choice.

Nonetheless, in response to the reviewer's comment, we fabricated new samples to conduct measurements to test relative enhancement in PL intensity. This was done by omitting the initial processing step of removal of a 10 nm InP capping layer by hydrochloric acid. As a result both sides of the new samples remain smooth, permitting pumping from the substrate under well controlled conditions. Therefore we could fairly compare the intensity of PL signals between the control InGaAsP MQW, a flat Ag/InGaAsP MQW interface, and nanostructured Ag/InGaAsP. (In the original experiments, pumping through the Ag layer and collection of emission through the Ag layer obscures the relative intensity enhancement due to uncontrolled scattering). Under the reverse excitation technique, we observed PL intensity enhancements in the LuHMS of ~350% and ~25%, relative to the control MQW and flat Ag/InGaAsP interface, respectively. A new supplementary section has been added detailing these results (Supplementary Section 7 in the revised SI).

They compare the results to that of plain QW. However the question is how does the emission pattern look when just a dielectric sub-wavelength grating (minus Ag) is used?

We have conducted additional measurements on control MQW samples that included subwavelength nanostructuring. The emission from these samples does not differ significantly from the unstructured control MQW. In the revised manuscript we have included a sentence about this comparison, and in the

revised SI we have added data on the nanostructured control MQW (last paragraph of Supplementary Section 2 and Fig. S6).

However this still does not address the main concern - what was gained by etching through quantum wells and putting metal right next to it?

Our response to this comment is identical to a similar comment of Reviewer 1. We repeat it for convenience:

“As discussed in [10], the spontaneous emission enhancement—Purcell factor and radiative enhancement—can be maximized when the light emitters are placed within the hyperbolic metamaterial (HMM). We have designed the LuHMS with this in mind. Emitters are distributed throughout the resulting HMM because the emitters and constituent dielectric are the same material. Conversely, other reported HMMs have incorporated light emitters in less systematic ways. For example, in [2] and [3] dye molecules were placed next to, but not within the HMM structure. In [4] a quantum dot was placed in a single location within the HMM, but not throughout the HMM as in our work. To more clearly explain the advantages of the LuHMS we have added reference [10] to the revised manuscript.”

Is polarization anisotropy at the expense of quantum yield the biggest gain?

The new experiment using the reverse excitation technique (Supplementary Section 7 in revised SI) shows that quantum yield is in fact increased when pumping from the substrate. Therefore, the “biggest gain” provided by the LuHMS ultimately depends on the application. If polarization anisotropy is desired, then a tradeoff may need to be made with quantum yield. However, if enhanced emission is desired, then this can be achieved at the expense of polarization anisotropy.

Furthermore, we have fabricated new samples with 3 nm films of SiO_x between the MQW and Ag layers to increase quantum yield by reducing quenching. Our additional experiments show that indeed PL intensity is increased by $\sim 1.25x$ in the flat and nanostructured Ag/InGaAsP samples with SiO_x , relative to those samples without SiO_x (Supplementary Section 7 in revised SI).

It appears that all that is happening is polarization filtering -can't that be done with pure dielectric structures?

The nanostructured InGaAsP MQW without deposited silver is a purely dielectric multilayer (air/InGaAsP). We do not experimentally observe polarization filtering, or any polarization anisotropy, with respect to the pump or emission in this structure (see Fig S6(a) and S6(b)). Additionally, we performed FDTD simulations of absorption and emission in InGaAsP nanostructures without Ag and found that polarization anisotropy was negligible. Therefore, we conclude that deeply subwavelength multilayers of dielectric (or semiconductor) *and metal* is necessary to achieve the effect of extreme (> 10 to 1 ratio) polarization anisotropy of pump and emission. The purely dielectric nanostructure will have an effect on the phase of incident and emitted light, but we did not study this theoretically or experimentally, and it therefore falls outside the scope of the present work

How bad did the quantum yield of the quantum well become?

When pumping and collecting emission through the Ag, the detected PL intensity from the metamaterial is $\sim 3\text{x}-10\text{x}$ smaller than the PL detected from the control MQW. However, when pumping and collecting from the substrate side, the detected PL intensity from the metamaterials is $\sim 3.5\text{x}$ greater than PL detected from control MQW.

In fact the same filtering effect is used to explain the blue shift in the emission.

We explain the blue shift in emission relative to the control MQW, regardless of pump polarization, due to the wavelength dependence of principal propagation of energy, as detailed in Supplementary Section 4. On the other hand, we explain the blue shift in emission in the LuHMS under parallel-polarized pumping, relative to normal-polarized pumping, as a consequence of greater absorption under the parallel-polarized pump. None of these effects would occur without the presence of nanostructured Ag, as confirmed by all of experiments and simulations.

The authors also carry out polarization anisotropy measurements on wavelength scale gratings and show that it is decreased when compared to LuHMS. They attribute this to the hyperbolic dispersion. However here again it is not clear what Ag does - will just a sub-wavelength grating do the job?

Again, the main idea of our manuscript is to demonstrate a hyperbolic metamaterial with emitter distributed throughout the structure. The grating couplers were designed and fabricated to show a controlled reduction in polarization anisotropy, and thereby provide further evidence for the existence of hyperbolic dispersion.

In summary, evidence for hyperbolic dispersion includes experimental demonstration of pump polarization anisotropy and emission anisotropy, reduction of pump absorption anisotropy and emission anisotropy via grating coupling, modification of the emission spectrum, and intensity enhancement relative to control MQW and flat Ag/MQW interface.

Simulations using effective medium approximation has been used to explain the observed effects.

In fact we conducted simulations using the effective medium approximation (EMA) and the exact nanostructure (see Fig. S7). We found that results between the two models were in good agreement, suggesting the utility of the EMA in designing the hyperbolic metasurface.

In summary I find the results to be of incremental nature, not convincing of the claims and most importantly limited in scope and of likely interest to a small sub-section of researchers working on hyperbolic metamaterials. The only key result is the polarization filtering and it is not even clear if one needs to go through all this hard fabrication to achieve the same goal!

Additional measurements show negligible difference between emission in control MQW with and without subwavelength nanostructuring. Therefore, the formation of multilayer Ag/InGaAsP MQW via deposition of Ag is required to achieve extreme polarization anisotropy. In the revised manuscript we have included mention of the similarity between structured and unstructured control MQW and have included additional data in the SI supporting this observation. Furthermore, we would like to add that

our fabrication procedure is, to date, the only demonstrated technique for incorporating gain media in hyperbolic metamaterials in the near-infrared (1200 nm -1600 nm) spectral window.

I recommend rejecting this manuscript.

We thank Reviewer 2 for his/her comments and suggestions.

Reviewer #3 (Remarks to the Author):

The manuscript presents an interesting new development in the field of active metamaterials - that combines the density of states super-singularity of the hyperbolic media with the simplicity of the fabrication offered by the metasurfaces. In future, this approach may eventually lead to a practical light-emitting device, as well as a new platform to study novel elements of light-matter interaction in artificial media. I therefore believe that the manuscript satisfies the stringent criteria of Nature Communications, and should be published.

However, as the two key elements in the proposed approach are (i) the super-singularity of the density of states in hyperbolic media, and (ii) the surface states with hyperbolic dispersion, the authors should reference the original papers that introduce these two concepts. To the best knowledge of the Referee, the photonic density of states super-singularity in hyperbolic media was first introduced in

I. I. Smolyaninov and E. E. Narimanov. Phys. Rev. Lett. 105, 067402 (2010),

while the first discussion of surface states with (in-plane) hyperbolic dispersion was presented in

Z Jacob and E.E. Narimanov, Applied Physics Letters 93 (22), 221109 (2008),

and the authors must reference these two papers (unless they find earlier work on the subject) in the introduction of the revised version of their manuscript.

We have added both of these references to the introduction of the revised manuscript. We thank Reviewer 3 for his/her comments and suggestions.

1. Z. Jacob, L. Alekseyev, and E. Narimanov, "Optical hyperlens: Far-field imaging beyond the diffraction limit," Opt. Exp. **14**, 8247–8256 (2006).
2. D. Lu, J. Kan, E. Fullerton, and Z. Liu, "Enhancing spontaneous emission rates of molecules using nanopatterned multilayer hyperbolic metamaterials," Nat. Nanotechnol. **9**, 48–53 (2014).
3. K. Sreekanth, K. Krishna, A. De Luca, and G. Strangi, "Large spontaneous emission rate enhancement in grating coupled hyperbolic metamaterials," Sci. Rep. **4**, 6340 (2014).
4. T. Galfsky, H. Krishnamoorthy, W. Newman, E. Narimanov, Z. Jacob, and V. Menon, "Active hyperbolic metamaterials: enhanced spontaneous emission and light extraction," Optica **2**, 62–65 (2015).
5. A. D. Neira, G. A. Wurtz, P. Ginzburg, and A. V Zayats, "Ultrafast all-optical modulation with hyperbolic metamaterial integrated in Si photonic circuitry.," Opt. Express **22**, 10987–94 (2014).
6. P. Segovia, G. Marino, A. V. Krasavin, N. Olivier, G. A. Wurtz, P. A. Belov, P. Ginzburg, and A. V. Zayats, "Hyperbolic metamaterial antenna for second-harmonic generation tomography," Opt. Express **23**, 30730 (2015).

7. Y. Sun, Z. Zheng, J. Cheng, G. Sun, and G. Qiao, "Highly efficient second harmonic generation in hyperbolic metamaterial slot waveguides with large phase matching tolerance.," *Opt. Express* **23**, 6370–8 (2015).
8. A. Salandrino, Y. Wang, and X. Zhang, "Nonlinear infrared plasmonic waveguide arrays," *Nano Res.* (2016).
9. Y. Guo and Z. Jacob, "Thermal hyperbolic metamaterials," *Opt. Express* **21**, 15014 (2013).
10. L. Ferrari, D. Lu, D. Lepage, and Z. Liu, "Enhanced spontaneous emission inside hyperbolic metamaterials," *Opt. Exp.* **22**, 4301–4306 (2014).

REVIEWERS' COMMENTS:

Reviewer #1 (Remarks to the Author):

I am happy with the responses and modifications to the publication and so recommend publication.

Reviewer #2 (Remarks to the Author):

In the revised version of the manuscript the authors demonstrated the polarization anisotropy and the spontaneous emission enhancement caused by the luminescent hyperbolic metasurface. The quality of the work is good. However it is an incremental work when one looks at the large body of work related to enhancement of spontaneous emission from hyperbolic media. Besides the new structure, what is the new science?

This work is more suited for a discipline specific journal.

Reviewer #3 (Remarks to the Author):

In the revised version of their manuscript, the authors have addressed all the major issues raised by the referees, and the paper can now be published.

The main criticism of the manuscript appears to come from the Referee 2, as the Referee 1 and myself only recommended minor changes that were straightforward to implement.

It appears that in his/her criticism of the manuscript, Referee 2 missed the major point of the work: that it presents the very first demonstration of the luminescent hyperbolic metasurface. As such, this work is new and nontrivial, and with a real potential for practical applications in the future. On this basis alone, it merits publication in Nature Communications.

Whether as a light source the luminescent hyperbolic metasurface is superior to other light emitters (such as e.g. a grating on top of a quantum well) right now, is therefore irrelevant - and any criticism based on this argument would be similar to rejecting the manuscript on the first transistor on the basis of the fact that its gain is inferior to that of the top-of-the-line vacuum tube.

Referee 2 also raised the question of whether the observed emission enhancement truly originates from the density of states super-singularity in the hyperbolic media, or has other origins. This is indeed an important and relevant question, and it has been fully addressed in the revised manuscript, with new samples and additional data.

I therefore recommend the publication of the manuscript without further delay.